# Tau mRNA Metabolism in Neurodegenerative Diseases: A Tangle Journey

**DOI:** 10.3390/biomedicines10020241

**Published:** 2022-01-23

**Authors:** Paulo J. da Costa, Malika Hamdane, Luc Buée, Franck Martin

**Affiliations:** 1Institut de Biologie Moléculaire et Cellulaire, “Architecture et Réactivité de l’ARN” CNRS UPR9002, Université de Strasbourg, 2, Allée Konrad Roentgen, F-67084 Strasbourg, France; pjgomes@unistra.fr; 2Inserm, CHU Lille, UMR-S 1172, “Alzheimer Tauopathies”, Labex DISTALZ, University Lille, Bâtiment Biserte, 1, Place de Verdun, F-59045 Lille, France; malika.hamdane@inserm.fr (M.H.); luc.buee@inserm.fr (L.B.)

**Keywords:** tau protein, neurodegenerative diseases, mRNA metabolism, translation

## Abstract

Tau proteins are known to be mainly involved in regulation of microtubule dynamics. Besides this function, which is critical for axonal transport and signal transduction, tau proteins also have other roles in neurons. Moreover, tau proteins are turned into aggregates and consequently trigger many neurodegenerative diseases termed tauopathies, of which Alzheimer’s disease (AD) is the figurehead. Such pathological aggregation processes are critical for the onset of these diseases. Among the various causes of tau protein pathogenicity, abnormal tau mRNA metabolism, expression and dysregulation of tau post-translational modifications are critical steps. Moreover, the relevance of tau function to general mRNA metabolism has been highlighted recently in tauopathies. In this review, we mainly focus on how mRNA metabolism impacts the onset and development of tauopathies. Thus, we intend to portray how mRNA metabolism of, or mediated by, tau is associated with neurodegenerative diseases.

## 1. Tau Protein and Associated Pathologies

In 1975, the tau protein was first isolated and recognized as a microtubule-associated protein (MAP) [1]. In fact, tau is a MAP that is highly expressed in neurons, predominantly in the axons [1,2,3,4]. Furthermore, tau binds to the outer microtubule surface tethering together tubulin dimers (α and β), stabilizing polymerization by a kiss-and-hop mechanism. Moreover, this mechanism was found to be essential for axonal transport in neurons [5,6,7].

Besides its primary function as microtubule dynamics regulator, tau also has roles in chromatin structure, signal transduction, synaptic plasticity, neuronal stress response and nucleic acid protection [8,9,10,11,12]. Considering this wide range of biological functions, it was surprising that pioneer reports on tau knockout mice (Mapt^-/-^) presented ‘physiological’ phenotypes with normal cognitive function in spatial learning tasks and presynaptic-mediated transmission [13,14]. However, more recently, Mapt^-/-^ mice exhibited age-dependent short-term memory deficits, hyperactivity, and synaptic plasticity defects. Thus, although tau has a role in normal neuronal function, its absence or reduction can be surpassed under certain conditions [15]. Nowadays, it is well established that tau KO animals display abnormal phenotypes [16]. In humans, the high relevance of tau goes beyond its physiological function. Indeed, abnormal tau aggregates are the hallmarks of several diseases/disorders classified under the general term of tauopathies.

Tauopathies comprise a group of proteinopathies of the human nervous system characterized by the assembly into filaments of tau proteins, alone or in association with other proteins. Among tauopathies are pathologies such as Alzheimer’s disease (AD), Pick’s disease (PiD), chronic traumatic encephalopathies (CTE), progressive supranuclear palsy (PSP), corticobasal degeneration (CBD), argyrophilic grain disease (AGD), frontotemporal dementia and Parkinsonism linked to chromosome 17 (FTDP-17) and several other, rarer diseases [17,18]. Moreover, tau aggregates display distinct morphological and biochemical features between tauopathies. In the brains of AD patients, tau aggregates are named neurofibrillary tangles (NFTs). NFTs are composed of two distinguishable intracellular assemblies, the paired helical filaments (PHFs) and the straight filaments (SFs) [18]. Besides affecting microtubule dynamics, pathogenic forms of tau are reported to affect nuclear pore localization, function and nucleocytoplasmic trafficking [19,20], induce heterochromatin relaxation [21], promote DNA damage [22,23], alter RNA stability and RNA export [24,25,26], promote ribosome instability [27] and activate transposable elements [28,29]. 

## 2. Tau mRNA Metabolism

In humans, tau is encoded by a single gene named MAPT (Microtubule Associated Protein Tau) located at human chromosome 17q21.31. Owing to an inversion polymorphism, the MAPT locus is found as two major haplotypes, named H1 and H2. The haplotype effect is related to changes in tau mRNA expression and splicing [30]. The MAPT gene consists of 16 exons numbered from 0 to 14. In the adult human brain, six major tau isoforms that arise from the alternative mRNA splicing of exons 2, 3 and 10 are expressed. The MAPT gene codes for tau isoforms that all consist of four domains, the N-terminal domain, a proline-rich domain (PRD), a microtubule binding domain (MTBD) that contains microtubule binding repeats, and a C-terminal domain that is involved in the regulation of microtubule polymerization [17,31].

### 2.1. Splicing Variants

Considering its six isoforms in the human adult brain, tau protein can vary between 352 and 441 amino acids. The alternatively spliced exons 2 and 3 code for 29 amino acids each, located in the N-terminal part of the protein (Figure 1A). The presence of exon 2 is mandatory for the inclusion of exon 3. Regarding exon 2 and 3 splicing, isoforms are named 0N (absence of both), 1N (exon 2 inclusion) and 2N (inclusion of both). Additionally, it has been reported that in the human brain, the 0N, 1N and 2N isoforms are differentially expressed, comprising 37%, 54% and 9%, respectively [32]. Furthermore, the alternative splicing of exons 2 and 3 can also affect tau protein interactions, as exon 3 inclusion is associated with decreased tau assembly into filaments [33,34]. Accordingly, the MAPT H1 haplotype that is known to be associated with tauopathies decreases exon 3 inclusion [35]. 

Alternative splicing of tau mRNA can also vary in its MTBD constitution, by either inclusion or exclusion of exon 10. Exon 10 encodes a 30-amino acid microtubule binding repeat. Consequently, the MTBD of tau can be composed of three or four microtubule binding repeats (3R or 4R). Given that 4R tau isoforms display increased MT-binding affinity and MT assembly when compared to 3R tau isoforms, the exon 10 alternative splicing has been reported to be crucial for the onset of neurodegenerative diseases [17,36,37]. Furthermore, it is known that the 3R/4R isoform ratio is a key factor for several tauopathies. Indeed, tau-mediated aggregates are heterogeneous and neurodegenerative diseases that can be divided into 3R, 4R or 3R + 4R tauopathies according to the biochemical composition of tau inclusions. Moreover, several mutations in and around exon 10 were found to affect its splicing and consequently linked to tau pathogenesis (detailed below) [17,18]. In this sense, AD and CTE are classified as 3R + 4R tauopathies; PiD and some types of FTDP-17 are 3R tauopathies; the 4R tauopathies include AGD, CBD and PSP [17,18,38]. Recently, Bachmann and colleagues (2021) showed that in human SH-SY5Y neuroblastoma cells, 2N tau isoforms diminished axonal enrichment while 4R isoforms increased microtubule count. These results point to the different but equally important consequences of alternative splicing of exons 2, 3 and 10 [39]. Thus, the alternative splicing of tau mRNA was proven to be crucial for tau-mediated pathology. Together with splicing deregulation, tau-mediated pathology is also dependent on the total amount of the protein [40,41]. Additionally, tau mRNA variants can arise from transcription initiation from different promoters and likely contribute to pathological processes.

### 2.2. Tau Transcription and Variable 5′ UTR 

A promoter is a genomic sequence located upstream of the transcriptional start site (TSS) that defines where the transcription starts. Consequently, a promoter determines the 5′ end of an mRNA. The existence of multiple transcripts with different 5′ ends originated from the same gene reflects the presence of alternative promoters in the genome. Moreover, the promoter usage may change according to tissues or cell types [42]. The alternative promoter usage (APU) contributes to transcriptome diversity and flexibility and the regulation of gene expression. Indeed, APU has been described to affect alternative splicing, tissue and/or subcellular specificity and gene expression during development [43,44,45,46]. 

For several years, only one promoter, located upstream of exon 0, was described for the MAPT gene. Interestingly, a hypomethylated CpG island (a feature associated with promoter regions) was found 13 kb upstream of exon 1. In addition, this region harbors binding sites for a transcription factor and a histone marker. These features suggest the existence of an alternative promoter in that region, a hypothesis corroborated by reports of shorter MAPT mRNAs starting on exon 1 in genome databases [47,48,49]. Indeed, Huin and colleagues (2017) reported a functional alternative promoter located in exon 1 of the MAPT gene that originates 5′ truncated transcripts in human brain (Figure 1B). These 5′ truncated transcripts could be translated into normal or N-terminal-truncated tau proteins, which could lead to changes in protein localization and function. In the brain, up to 60% of the genes are transcribed by APU [44,50]. Moreover, the usage of different promoters has been described for genes involved in Alzheimer’s disease, such as apolipoprotein E gene (APOE) and the presenilin-2 gene (PSEN2) [50,51,52]. Even though the activity of the MAPT promoter located in exon 1 is weak, it was found to be relevant for tissue-specific expression [53,54]. Interestingly, the 5′ end truncated variants of tau mRNAs were found to be overexpressed in AD and PSP brains [48]. Notably, as depicted in Figure 1B, several 5′ end truncated mRNAs were found in AD patient brains (−160, −54, −47, −35, −7, +90, +160 and +201) and absent from healthy control brains. Once the 5′ end truncated transcripts can lead to 5′ terminal truncated proteins, this can be relevant for the onset of neurodegenerative diseases. Tau truncated protein expression was associated with neurodegenerative diseases in mice and humans [55,56,57,58,59]. Among the 5′ truncated tau mRNAs were particularly interesting transcripts with the first AUG codon located in the 48th position of the coding sequence (Met11 in the protein) or in exon 5 (Met127 in the protein) (Figure 1B). Interestingly, these N-terminally truncated tau proteins starting from Met11 and Met127 had already been detected in human brain tissue using a proteomic approach [60]. In the long 5′ UTR of tau mRNA, the existence of an internal ribosome entry site (IRES) was unveiled [61,62]. Initially characterized in viral genomes, an IRES is a sequence of mRNA in the 5′ UTR that can initiate translation without the need for all translation initiation factors [63]. In fact, the IRESs fold into a secondary structure that is able to directly recruit the 40S ribosome in a cap-independent manner. Cellular IRESs were shown to work as a by-pass mechanism when canonical translation is inhibited [64]. The IRES in tau mRNA is defined by two main structural domains and is very sensitive to small nucleotide substitutions. Additionally, the tau IRES located between 230 and 142 nt upstream of the canonical AUG was able to directly recruit the 40S ribosome (Figure 1B) [61,62]. Interestingly, the IRES is present only in tau mRNAs with a long 5′ UTR, thereby raising questions about the translation initiation mechanism used for tau mRNA variants with short 5′ UTRs.

In fact, APU diversifies the transcriptional profile not only by the different 5′ ends but also by, among others, modulating alternative splicing, tissue specificity and/or expression levels [43,65,66,67]. In addition, the existence of APU allows the promoter switching that could lead to a harmful process or to better responses to diverse stimuli. As previously described, these 5′ end truncated transcripts of tau could also drive different splicing isoforms. Even though the tau second promoter function has yet to be fully described, it may alter the 3R/4R tau protein ratio and possibly drive tau pathology [48]. This report suggests that beyond the protein truncation driven by post-translational proteolysis, APU can also generate N-terminally truncated tau variants. Therefore, APU can represent an underestimated source of tau protein variants that could have an impact in tauopathies. 

### 2.3. The 3′ UTR Is Important for Tau mRNA Trafficking 

Despite being controversial, it was reported that 3′ UTR of tau mRNA could be essential for its axonal localization and, consequently, tau function [68]. In their report, Aronov and colleagues stated that absence or mutation in what they called 3′ UTR axonal targeting region of tau mRNA resulted in mislocalization of tau mRNA and protein in neuronal cells. Furthermore, swapping the tau 3′ UTR with the dendrite targeting 3′ UTR of MAP 2 (microtubule-associated protein 2 that localizes to cell body and dendrites) caused tau mRNA and protein to fail to localize to axons [68]. On the other hand, dendritic microinjections of tau protein or neuron transfection of tau cDNA without tau 3′ UTR also localized in axons [69,70]. More recently, tau proper localization was linked to autophagy and proteasomal pathways rather than mRNA localization [71]. Moreover, there are additional hypotheses for the regulation of tau protein sorting such as annexin, which interacts with tau exon 1 and thereby contributes to tau axonal localization [72]. In addition, it has been reported that miR-132 microRNA directly targets tau 3′ UTR [73,74]. Remarkably, the activity of this miRNA can impact tau aggregation, cognitive function, and tau metabolism [74]. The link and relevance of noncoding RNAs to tau post-transcriptional and post-translational regulation will be explored below.

### 2.4. Noncoding RNAs: From microRNAs to Long Noncoding RNAs

Noncoding RNAs (ncRNAs) are a class of RNAs that are transcribed like mRNAs but do not code for functional proteins. Instead, ncRNAs are known to play crucial regulatory roles in mRNA and protein metabolism. The ncRNA class includes long noncoding RNAs (lncRNAs), piwi-interacting RNAs (piRNAs) and microRNAs (miRNAs) [75,76].

MicroRNAs (miRNAs) are small RNAs (~22 nts long) that regulate the expression of complementary messenger RNAs and, therefore, play important posttranscriptional regulatory roles in eukaryotic cells [77]. In humans and other mammals, these small RNAs help modulate the expression of most mRNAs shaping the transcriptome [77]. In brief, miRNAs are transcribed by RNA polymerase II as primary miRNAs (pri-miRNAs). The pri-miRNAs are processed by Drosha/DGRC8 complex in the nucleus before being exported to cytoplasm as a hairpin-shaped precursor miRNA (pre-miRNA). Once in the cytoplasm, Dicer generates a ~22 nt long miRNA duplex [77]. Then, one strand of the miRNA duplex (the mature miRNA) in association with the RNA-silencing complex (RISC) will encounter the complementary target mRNA, leading to its translational repression or degradation. The miRNA-mediated regulation is very versatile, since a single miRNA can target several mRNAs and a single mRNA can be under the control of different miRNAs [77]. Due to the versatile nature of miRNAs, it is not surprising that miRNAs are involved in several brain functions such as neurogenesis, memory, synaptic function, and neuronal survival [78,79]. In this context, several miRNAs were indicated as good biomarkers for some neurodegenerative diseases, specifically AD. In fact, several miRNAs were found to be differentially expressed in the brain, serum and plasma of AD patients [73,80,81]. Interestingly, several miRNAs were directly or indirectly involved in tau pathology. Among them, miR-125b, which is up-regulated in AD, promotes tau hyper-phosphorylation in neuronal cells. This indirect effect arises most likely through down-regulation of the phosphatases DUSP6 and PPP1CA, which are considerably reduced in AD brains [82]. Corroborating this hypothesis, suppression of miR-125b in neurons reduces tau phosphorylation and kinase expression/activity. Conversely, miR-125b overexpression impairs associative learning and downregulates DUSP6 and PPP1CA, resulting in enhanced tau phosphorylation in mice. These results show that miR-125b has a role in the tau pathology in AD [82]. Additionally, dysregulation of miR-146a was also reported to induce tau hyperphosphorylation and AD pathogenesis via ROCK1-PTEN-tau pathway [83]. Interestingly, deficiency in the miR-132/212 cluster, which is downregulated in tauopathies, leads to tau overexpression, phosphorylation, and aggregation. The miR-132 is very versatile and, besides directly targeting tau mRNA in its 3′ UTR, is predicted to target several other proteins relevant to neuronal survival and synaptic function [73,74]. Furthermore, miR-132/212 deletion induced tau aggregation in mice expressing endogenous or mutant tau. Treatment of AD mice with miR-132 mimics partially restored memory function and tau metabolism. Interestingly, miR-132 and miR-212 levels correlate with insoluble tau filaments and cognitive impairment in humans [74]. Moreover, miR-132 was also implicated in the regulation of tau alternative splicing, which is dysregulated in PSP [84]. However, miR-132 is not the only miRNA modulating tau alternative splicing. MiR-124, miR-9, miR-137, and miR-153 were also reported to affect the alternative spicing of tau exon 10 in neuronal cells by targeting splicing factors such as poly-pyrimidine tract-binding proteins 1 and 2 (PTBP1/2) [74,85,86]. The miR-132/232, among other miRNAs, such as mir-9 and miR-181c, was also reported to directly inhibit the deacetylase SIRT1 [87]. In AD brains, reduced SIRT1 levels increased acetylated tau levels and pathological phosphorylated tau in vivo [88]. Interestingly, miR-132-3p and miR-22-3p shared 48 target genes, such as PTEN and SIRT1, which directly or indirectly impact mechanisms associated with tau pathology [73].

Besides microRNAs, another noncoding class of RNAs comes into play in the regulation of tau—the long noncoding RNAs (lncRNAs). LncRNAs are commonly defined as transcripts longer than 200 nucleotides that are not translated into proteins [75,76]. Lately, an increasing number of reports have described and characterized the function of lncRNAs. Broadly, lncRNAs can be categorized as *cis* regulators and *trans* regulators. The former regulate chromatin structure and/or gene expression near the transcription site. The latter perform cellular functions throughout the cell far from the transcription site. Furthermore, lncRNAs can be generated by pervasive transcription mostly in the opposite direction of the transcript that they regulate (*cis* regulation) [76]. Of note in the scope of this review, an lncRNA was described and characterized for an MAPT locus, the MAPT-AS1. MAPT-AS1 is an 840 nt lncRNA transcribed from the anti-sense strand of the MAPT promoter region [89,90]. Moreover, MAPT-AS1 expression was reported to be reduced in brains of patients with Parkinson’s disease [89,90]. MAPT-AS1 is upregulated during neuronal differentiation [91]. Additionally, while overexpression of MAPT-AS1 leads to a decrease in MAPT promoter activity, MAPT-AS1 knockdown increased methylation and 4R isoform expression (but not of total tau transcripts) in cellular assays. This suggests that MAPT-AS1 has a role in both gene transcription and alternative splicing of tau [89].

More recently, MAPT-AS1 was further characterized as a functional expression regulator of neuronal tau [92]. Indeed, this report showed that the 75-nt overlap with the tau mRNA in its 5′ UTR modulates neuronal tau protein levels and tau pathology in human brain [92]. Moreover, MAPT-AS1 as well as tau is enriched in the brain, being found in both nucleus and cytoplasm. Further, it was proposed that MAPT-AS1 mediates repression of tau translation by competing for ribosomal-RNA pairing with tau mRNA IRES [62,92]. Altogether, these results define a role of MAPT-AS1 in regulating tau translation as an important player in the prevention of tau-mediated pathology.

## 3. Tau Protein Metabolism

All Tau isoforms comprises four domains: N-terminal domain (aa 1–150), proline-rich domain (PRD), microtubule binding domain (MTBD) and C-terminal domain (aa 369–441) (Figure 2) [18]. The tau N-terminal domain, also known as projection domain, plays a role in signaling and protein interactions, acting as function regulator and contributing to tau’s physiological paperclip conformation by interacting with the C-terminal domain. The so-called projection domain projects away from microtubules when tau binds to microtubules [31,93]. Tau N-terminal domain contributes to its axonal localization and has been proposed to mediate interactions with neuronal proteins. Further, N-terminally truncated tau showed increased association with microtubules [72,94].

The PRD (aa 151–243) is responsible for interaction with proteins containing SH3 domains and plays an important role in tau protein folding. Furthermore, the PRD plays a role in cell signaling and interacts with protein kinases. Consequently, the PRD impacts not only microtubule binding and assembly, but also tau phosphorylation sites [31,93,95].

The MTDB domain (aa 244–368) mediates the tau binding to microtubules and is responsible for microtubule stabilization [17,31].

The C-terminal part (aa 369–441), in addition to contributing to microtubule binding, interacts with the N-terminal part conferring the paperclip conformation to tau protein [17,31].

### 3.1. MAPT Mutations of Tau Proteins

To date, more than 100 MAPT mutations have been associated with neurodegenerative diseases [96] (https://www.alzforum.org/mutations/mapt) (accessed on 17 January 2022). These mutations can be divided into two groups, missense mutations or splicing mutations. The primary effect of the missense mutations is at the protein level, while splicing mutations modulate the alternative splicing of tau exon 10. Mutations that introduce amino acid substitution have an impact by altering the tau interactions with microtubules promoting tau assembly into filaments [97,98,99]. 

Besides affecting the mRNA splicing of MAPT exon 10, these mutations can disrupt tau protein folding and/or protein–protein interactions, enhancing tau aggregation. Actually, most of the mutations are located between exon 9 and exon 12, which encodes for the MTBRs (Figure 2A). Thus, these mutations can affect the tau interaction with microtubules and promote tau assembly into filaments [18,96,97,98,99,100,101,102]. Remarkably, a direct correlation between MAPT mutations and post-translational modifications of tau has not been established [102]. 

As depicted in Figure 2, most of the MAPT missense mutations cluster in or near the MTBR domain of tau as well as the splicing mutations. The mutations that have an impact on splicing of exon 10 can be either intronic or exonic. These mutations usually result in overproduction of 4R tau and its assembly into filaments [17,103]. Although less common, it has been shown in several reports that mutations can also increase 3R tau expression. Indeed, mutations in exons 9–13 (K257T, L266V, G272V, G273R, S305N, L315R, S320F, S320Y, P332S, Q336H, Q336R, K369I, E372G and G389) were associated with 3R tauopathies, such as PiD [17,104] (https://www.alzforum.org/mutations/mapt) (accessed on 17 January 2022). Recently, a deletion from G389 to I392 was also described as predominant in 3R tauopathy [105]. However, the most characteristic MAPT mutations affect the alternative splicing by inclusion of exon 10, resulting in the overproduction of 4R Tau. In fact, several intronic mutations (I9-10 G > T, I9-15 > C, I10+3 G > A, I10+4 A > C, I10+11 T > C, I10+12 C > T, I10+13 A > G, I10+14 C > T, I10+15 A > C, I10+16 C > T, I10+19 C > G, I10+25 C > T, I10+29 G > A) and exonic mutations (N279K, ΔK280, L284L, L284R, C291R, ΔN296, N296D, N296H, N296N, K298E, G303V, G304S, S305I, S305N, S305S and V363I) were associated with neurodegenerative diseases, often associated with 4R overproduction [17,103,104,106,107,108]. In addition, the P301 residue, present only in 4R tau isoforms, is of major importance for regulation of tau aggregation [103]. Indeed, mutations on P301 are highly relevant for studying tauopathies mainly because P301L is the most common tau mutation. Furthermore, in transgenic mice, the P301L mutation induces aberrant tau hyperphosphorylation and aggregation [20,109]. Additionally, there are three mutations on this residue (P301L, P301S, P301T) associated with tau pathology [110]. Indeed, it was shown that P301L, P301T, and P301S recombinant proteins are prone to induce more oligomers when compared with wild-type recombinant tau [110]. In addition, a PS19 mouse model that bears a P301S tau mutation spontaneously develops tau pathology [111]. In fact, tau transgenic mice expressing P301L or P301S human tau develop tau-mediated pathology [111,112,113,114]. Additionally, the combination of P301L or P301S tau with S320F generated aggressive models of tauopathy without exogenous seeding [115]. Furthermore, THY-Tau 22 mice (bearing two mutations on tau G272V and P301S) show hyperphosphorylation of tau, NFT-like inclusions and cognitive impairment [116]. However, MAPT mutations other than P301 also enable the generation of in vivo models of tau pathology.

Mutations outside exon 10, such as I260V in exon 9, K317M and K317N in exon 11, E342V in exon 12 and N410H in exon 13, were also able to alter the 3R/4R ratio [17]. Nevertheless, not all mutations located in exons 11 and 12 affect exon 10 splicing like G335S, G335V, G335A and V337M. Instead, they reduce the ability of tau to promote microtubule assembly [17,117]. Conversely, Q336H and Q336R mutations increased tau-mediated microtubule assembly [17,118]. Despite not altering the 3R/4R ratio or having the ability to bind MTs, the D348G tau mutation was associated with early amyotrophic lateral sclerosis (ALS) onset. For the D348G mutation, ALS seems to be caused by tau accumulation due to evasion of proteasomal degradation [119,120]. The S352L, S356T, P364S, G366R, P397S, R406W and T427M mutations were found in FTLD, FTD or PD patients [121,122,123,124,125,126]. Notably, the P364S tau showed a special propensity for aggregation, even higher than that of P301L tau. Moreover, P364 (in R4) corresponds to the residue P301 in R2 and to P322 in R3, whose mutations were strongly associated with different tauopathies [123]. It is also worth noting that more C-terminal mutations (P397S, R406W and T427M) were associated with later age of onset and/or longer disease duration, which suggests that the C-terminal part of the protein is less critical for tau-mediated pathology [124,125,126].

Interestingly, there is only one residue (R5) located in exon 1 in the N-terminal of the tau protein that is associated with tau-mediated pathology. The R5H, R5L and R5C mutations were associated with AD, PSP and Parkinson’s disease, respectively [127,128,129]. The R5H and R5L mutations reduce the ability of tau to promote microtubule assembly and promote tau fibril formation [128,130]. Furthermore, the R5L mutation was shown to affect tau polymerization and MT assembly in 0N4R isoforms [131].

### 3.2. Post-Translational Modifications of Tau Protein

Tau pathology is often explained by changes in its post-translational state, regarding for example, its phosphorylation and truncation pattern [132,133]. Indeed, tau protein can be subjected to several post-translational modifications, such as phosphorylation, acetylation, methylation, glycation, ubiquitination and truncation [132,133,134,135,136,137]. Tau post-translational modifications are tightly regulated and essential for its proper function and metabolism as well as its relevance for neurodegenerative disorders [135,137]. 

Tau phosphorylation is a normal event, occurring on serine, threonine and tyrosine residues. On average, in the normal adult brain, tau has 2–3 phosphorylated residues, whereas in the AD brain the phosphorylation is at least 3-fold greater. Interestingly, depending on the isoform, tau can have up to 85 putative phosphorylation sites. However, not all of the putative phosphorylation sites were described and/or characterized (Figure 2B) [93,135,138]. Usually, hyperphosphorylation impairs the ability of tau to interact with MTs and can induce tau missorting. Consequently, tau hyperphosphorylation can drastically alter the function of the protein and induce tau pathology [93,139]. In addition, the hyperphosphorylation of tau has been associated with aggregate formation and microtubule assembly disruption [31,103,140,141]. Thus, there are some therapeutic approaches focusing on serine/threonine dephosphorylation [93]. Although hyperphosphorylation is strongly associated with aggregate formation, there are phosphorylation sites that can have a protective effect against aggregation. In fact, several phosphorylation sites are shared by normal and AD brains, and some phosphorylation sites are exclusively found in normal brains (Figure 2B) [93,135]. For example, on one hand, the phosphorylation on S262 and S356 residues is AD-specific and decreases the binding affinity of tau to microtubules. On the other hand, S293 and S305 residues were found phosphorylated only in normal brains, which suggests that they might have a protective effect against aggregation (Figure 2B) [132].

Of note, it has been shown that phosphorylation at AT8 sites in tau (S199E+S202E+T205E) promotes the split of the N-terminal and C-terminal domains [135,142]. However, controversial data showing that this AT8 epitope actually corresponds to phosphorylated residues S202, T205 and S208 have also been presented [143]. In turn, phosphorylation at the PHF1 epitope (S396E+S404E) moves the C-terminal domain away from the MTBR domain. Interestingly, combining both phosphorylations leads to compaction of the paperclip conformation, which leads to the tau aggregation characteristic of tau in AD [142]. Furthermore, there is growing evidence that acetylation is the second most important PTM for the function of tauopathies after phosphorylation. Since the longest tau isoform (2N4R) has 44 lysine residues, up to 10 percent of the protein has the potential to be acetylated [137]. In fact, more than 20 lysine residues of tau are subjected to acetylation [88]. Interestingly, tau acetylation blocks its own degradation, inhibits microtubule binding and promotes aggregation [88,144,145,146]. Moreover, tau acetylation affects tau phosphorylation and ubiquitination [88,147]. Additionally, as mentioned above, the deacetylase SIRT1 directly targets tau, reducing tau acetylation and, consequently, tau accumulation [148]. Furthermore, the strong relevance of tau acetylation might be linked to the prevention of other modifications at tau lysine residues, including methylation, glycation and ubiquitination [137]. In addition to acetylation, lysine residues of tau can also be ubiquitinated. Ubiquitination is the most common PTM in mammalian cells and can affect the activity, localization or stability of the targeted proteins [149]. Strikingly, tau ubiquitination was reported to increase both tau clearance and tau aggregation [137,150]. Besides acetylation and ubiquitination, tau lysine residues can also undergo methylation and glycation. Both methylation and glycosylation can impair tau microtubule-binding activity, proper clearance and aggregation [137,151]. Thus, tau methylation and glycation also contribute to neurodegenerative disease pathogenesis [137]. Taken together, PTM combinations, competition and crosstalk add a new layer of complexity to the regulation of tau function. Notably, this seems particularly relevant at tau lysine residues. Besides these usual PTMs, tau can be subjected to proteolytic truncation (tau proteolysis). The resulting tau fragments are also recognized to drive tau pathology and contribute to dysregulation of tau function, as described in more detail below (Figure 2C) [72,152,153]

### 3.3. Tau Proteolysis 

For several years, it has been suggested that PTMs of tau induce protein structural changes that disrupt the physiological MT-binding paperclip conformation [93]. The MT release and consequent cytoplasmic accumulation allows tau proteolysis that ultimately can lead to tau aggregates [93,152]. Along with this process, tau proteolysis, also known as tau truncation, was found to promote tau aggregation [59]. Moreover, a wide range of tau fragments already have been found to be neurotoxic, contributing to the onset of tauopathies [140,152]. 

Tau protein cleavage can be classified by the cleavage site and responsible protease, as depicted in Figure 2C [140,152]. Several tau fragments were already attributed to several proteases. However, several proteases responsible for many of the tau fragments are still not characterized (Figure 2C) [140,152]. The relevance of proteases and the different tau fragments to tau pathology has been shown in several reports. For example, blocking caspase-2-mediated cleavage rescues memory deficits in P301L tau transgenic mice [154]. Furthermore, deletion of the first 150 or 230 amino acids of tau enhances tau self-aggregation and seeding. Conversely, loss of the 50 or 20 C-terminal amino acids does not produce significant effects [155]. Moreover, Gu and colleagues (2020) showed that the tau fragment from residues 151 to 391 has the highest pathological activity in vitro. Thus, these data indicate that some sections of tau can prevent tau aggregation and consequently protect tau from pathological features [155].

Tau truncation can be mediated by a wide variety of proteins. Indeed, several reports show that tau can be cleaved by ADAM10, calpains (calpain-1 and calpain-2), caspases (caspase-2, -3 and -6), asparagine endo peptidase, thrombin, chymotrypsin and cathepsins (cathepsin B, D and L). Besides this, tau can also be cleaved by acetylation-induced auto-proteolysis.

Amongst the caspases, caspase-2 was shown to produce the fragment 1-314, and this C-terminally truncated tau fragment promotes tau aggregation in dendritic spines [154]. Moreover, caspase-3 cleavage between D421-S422 was associated with AD and PSP. Additionally, caspase-6 produces a 1-402 tau fragment used as biomarker for AD [140,152].

The calpains that mediate tau proteolysis are calpain-1 and calpain-2. Calpain-1 has been associated with several neurodegenerative diseases such as Parkinson’s disease and Huntington’s disease. Moreover, a fragment generated by calpain-1, tau 232-441, was considered cytotoxic and associated with aging [140,152].

In conclusion, there are many different PTMs that affect tau and is functions. Ultimately, these PTMs, such as proteolysis, can induce tau aggregation and, consequently, tau pathology. Thus, the better our understanding of how tau PTMs are regulated, the sooner we will have in our hands tools to decipher the disease mechanisms of tauopathies and consequently design new therapeutic strategies.

## 4. Tau Protein and Cellular mRNA Metabolism

The longstanding relation between tau and RNA was established by the ability of RNA to act as a cofactor for tau aggregation [156,157]. The negatively charged nature of RNA neutralizes the positively charged residues of tau [156,157]. Interestingly, an increasing number of reports have shown that mutations in genes that encode RNA binding proteins cause neurodegenerative diseases such as ALS, FTLD, AD and spinal muscular atrophy (SMA) [158,159,160]. Interestingly, some interactome studies unveiled that tau protein interacts with RNA binding proteins such as TIA1, TDP-43, DDX6, PABP and RPL7 [160,161,162,163].

The link between the RBP T-cell intracellular antigen 1 (TIA1) and tau appears to be of major importance, since TIA1 is a core component of stress granules (SGs). SGs are cytoplasmic membraneless ribonucleoprotein complexes that under stressful conditions reversibly aggregate in the cytoplasm. SGs can arise in response to diverse cellular stresses such as nutritional stress, heat or osmotic shock, DNA damage or proteostasis dysfunction [164,165,166,167,168,169,170]. Indeed, under different stresses, the nuclear TIA1 is translocated to the cytoplasm, where it becomes one of the main building blocks of SGs [166]. Additionally, SGs are known to be constituted by mRNAs, RNA binding proteins, the small 40S ribosome subunit and translation factors [170,171,172]. Moreover, SGs can regulate mRNA metabolism and protein translation [173]. Concerning tauopathies, TIA1 reduction was able to prolong the lifetimes and increased neuronal survival in transgenic P301S tau mice [162]. Interestingly, under stress triggered by arsenite, glucocorticoids, Aβ and tau oligomers, the TIA1 and hyperphosphorylated tau co-localize in the SGs [174,175,176,177]. Even though SGs are not homogeneous, hyperphosphorylated tau inclusions strongly colocalize with TIA1-containing granules in the neuronal SH-SY5Y cell line [175,178]. Furthermore, in tau KO neurons, TIA1 is strictly nuclear, which hints that tau may have a role in TIA1 translocation from the nucleus [177]. Moreover, Tau/TIA1 co-localization was correlated with larger tau aggregates [176]. More recently, it was shown that reducing TIA1 in vivo rescued the tau-associated phenotype in P301S tau mice [162,179]. Even upon tau overexpression, reducing TIA1 levels not only decreased the number and size of cytoplasmic TIA1-containing granules, but also rescued behavior, neuronal degeneration and survival [162,179]. Similarly, TIA reduction decreased the hyperphosphorylated tau accumulation in 3-month-old mice [162,179]. In accordance with this observation, it has been shown in vitro that TIA1 knockdown also reduces tau pathology and provides neuroprotection [177]. Recently, Ash and colleagues showed that tau oligomerization in association with TIA1 alone is more toxic than tau aggregates generated by incubation with RNA alone or artificial crowding agent polyethylene glycol (PEG) [12]. 

Besides TIA1, pathological tau (hyperphosphorylated or misfolded) was found to co-localize with multiple RNA binding proteins [161,162,175,176,177,179]. These studies raise the hypothesis that oligomeric or misfolded tau is cytotoxic and able to drive cognitive decline independently of NFT formation [180,181]. Moreover, it was reported that RNA splicing is dysregulated in AD brains and in a PS19 mouse model (P301S tau). In the PS19 brain, transcripts encoding spliceosomal and mRNA binding proteins are reduced, favoring transcripts that encode synaptic proteins. Interestingly, reducing TIA1 expression in PS19 mice partially rescued the “physiological” RNA splicing [179]. 

Even though the tau association with SGs has not been thoroughly described, the link between tau and RNA metabolism is emphasized by the direct interaction of phosphorylated tau with TDP-43 [159,160]. TDP-43 is an RNA and DNA binding protein mainly associated with mRNA metabolism [182,183,184]. TDP-43 is mainly nuclear but can also be found in cytoplasm [185,186], where it is known to incorporate SGs [186,187]. As part of its nuclear function, TDP-43 is involved in transcription, translation, mRNA transport, microRNA biogenesis and long non-coding RNA processing [186,188]. Interestingly, it was shown that tau/TDP-43 interactions might be involved in several tauopathies by affecting each other’s functions. On one hand, TDP-43 oligomers may serve as nucleation sites for tau aggregates; on the other hand, tau oligomers modulate the TDP-43 aggregation state and cellular localization [160]. 

Interestingly, TDP-43 also directly interacts with TIA1 and is detected in SGs of FTD and ALS patients [160,189,190]. Furthermore, TDP-43 has been implicated in several neurodegenerative diseases, such as FTLD, ALS, FTD and AD [160,191,192]. It has been suggested that tau association with TDP-43 oligomers might play a role in AD, ALS and FTD [160]. Although TDP-43 aggregates may modulate AD pathology, there are AD cases without TDP-43 pathology [193,194]. Even though not strictly required, these reports suggest that TDP-43/tau interaction in SGs might be critical for several tauopathies and neuronal function.

Still in the context of SGs, tau also interacts with the splicing factor proline and glutamine rich (SFPQ) protein. SFPQ is an abundant nuclear RBP, being a key component of neuronal transport granules. SFPQ is involved in various processes such as alternative splicing, DNA repair, transcriptional regulation and RNA processing and transport [195,196]. SFPQ is dysregulated in association with tau and TIA1 proteins in patients with rapidly progressive AD. This dysregulation has been associated with the nuclear depletion and cytoplasmic co-localization with TIA1 and tau proteins [197].

More recently, Jiang and colleagues (2021) showed that hnRNPA2B1 protein is required for tau-mediated neurodegeneration in vitro and in vivo [198]. The hnRNPA2B1 protein is a reader of m^6^A in the nucleus, where it regulates the chromatin state and transcription [199,200,201]. Moreover, the oligomeric tau (oTau) complexes with hnRNPA2B1 and m^6^A-containing RNA seem to regulate RNA translation stress response and promote SG formation. It was suggested that hnRNPA2B1 functions as an m^6^A reader bridging oTau to m^6^A labeled transcripts. Interestingly, hnRNPA2B1 knockdown reduces the toxicity induced by pathological tau and dampens the stress response in neurons [198].

Beyond its involvement in SGs, the link between tau and RNA translation regulation is strengthened by tau acting as a negative regulator for protein synthesis by binding to ribosomes [27]. Indeed, tau can function as translation inhibitor by sequestering ribosomal protein S6 (rpS6), a component of the 40S ribosome complex, which is essential for canonical translation [202]. Moreover, human AD brains showed decreased rpS6-dependent translation, establishing tau as a selective driver for global mRNA translation [202]. This tau–ribosome interaction could help to explain the link between tauopathies and cognitive impairment [202,203]. 

Moreover, tau oligomers co-localize with Musashi proteins (MSI1 and MSI2) in AD, ALS and FTD brains [204]. MSI proteins are RNA-binding proteins that regulate mRNA translation during neuronal development in *Drosophila melanogaster* [205]. MSI1 can be found in both cytoplasm and nucleus, while MSI2 was found in the cytoplasm and associated with polysomes [206]. Tau and MSI proteins co-localize in the cytoplasm and nuclei of mouse neurons [204]. Tau regulates MSI function by modulating MSI levels and controlling their cellular localization and interactome [204]. 

Recently, an additional, direct link between tau and mRNA translation inhibition was found. Two reports revealed a direct interaction between tau and DDX6 [163,207]. DDX6 is a DEAD-box RNA helicase known as mediator of mRNA decay and miRNA biogenesis [208]. The interaction between tau and DDX6 increased the silencing activity of let-7a, miR-21 and miR-124 miRNAs. Indeed, it was shown that the tau/DDX6 interaction potentiates let-7 miRNA silencing activity rather than affecting the translational repression mediated by DDX6. Furthermore, disease-associated mutations of tau P301S and P301L disrupted tau/DDX6 interaction and consequently impaired let-7 mi-RISC-mediated gene silencing. Altogether, these data reveal a new role for tau in regulating miRNA activity [163]. Furthermore, DDX6 accumulates in AD and CBD human brains, where it co-localizes with hyperphosphorylated tau [163]. 

Interestingly, it was described that tau also has a nuclear role, being crucial for DNA and RNA integrity in vivo. Furthermore, tau oligomerization triggers the loss of the nucleic acid protective effect of monomeric tau [10,23]. Besides that, tau has a major impact on transcription and post-transcription regulation. Indeed, tau nuclear levels and phosphorylation state reduced global synthesis of RNA (including rRNA) and was suggested to be relevant to ribosomal biogenesis and function [209,210]. Moreover, tau was shown to be crucial for ribosomal DNA transcriptional repression through its involvement in heterochromatin and DNA methylation [210]. In addition, tau seems to have a role in translocating nuclear RBPs to cytoplasm [176,177]. This might be due to tau interaction with nucleoporins impairing protein transport between the nucleus and the cytoplasm [20]. Considering that under basal conditions most RBPs are nuclear, under stress, they are translocated to the cytoplasm, where they may trigger SG formation [161,211]. Although the precise mechanism by which tau modulates transcription is unknown, tau involvement in heterochromatin and DNA methylation is a factor to take into consideration. Moreover, tau-mediated translocation of known transcription regulators such as SFPQ and TDP-43 to the cytoplasm might be of utmost importance. Thus, tau aggregation into SGs may also be a key factor for tau transcription modulation. The relevance of tau in this process grants tau a pivotal role in global mRNA metabolism. Altogether, SG formation and protein aggregation have a huge impact on tauopathies due to their role in transcription regulation, RNA integrity, ribosomal function and subcellular compartmentalization. Finally, these studies show that tau-mediated RNA metabolism can be a powerful driver of neurodegenerative diseases. However, whether tau-mediated RNA metabolism alone can drive tau pathology is still an open question.

## 5. Conclusions

The molecular mechanisms underlying the onset and development of tauopathies are very complex. The undeniable role of tau post-translational modifications has been extensively studied. In this review, we highlighted the interest in investigating not only the role of tau protein in general mRNA metabolism but also the role of tau mRNA metabolism per se in the development of tauopathies. The diversity of tau mRNA variants impacts tau protein properties and its physiological and pathological functions. This diversity is mainly related to alternative splicing, but other variables such as the alternative promoter usage (APU) or the involvement in the general mRNA metabolism (SG formation) may also contribute and need to be further investigated. Likewise, APU could have an impact on tau translation and the production of some N-terminal truncated tau species whose role in tauopathies is well-known. Therefore, some aspects of tau mRNA metabolism are likely among the mechanisms underlying translation deregulation and able to play an instrumental role in the physiopathology of tauopathies.

## Figures and Tables

**Figure 1 biomedicines-10-00241-f001:**
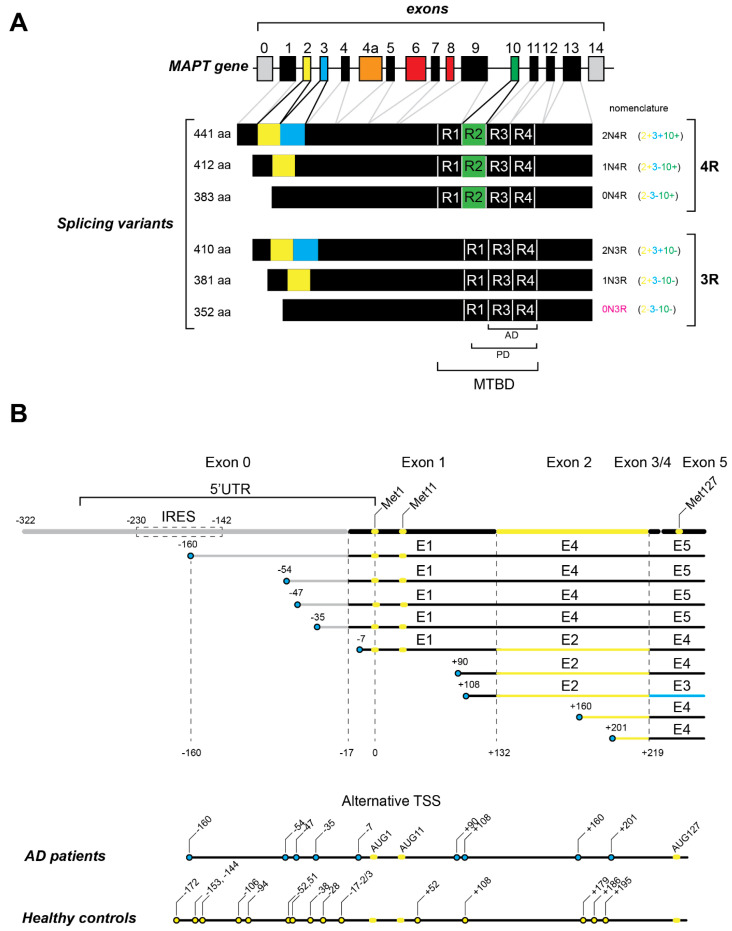
Tau mRNA transcripts. (**A**) Tau mRNA transcripts are generated from the single MAPT gene and consist of 16 exons (numbered 0 to 14). The coding sequence starts in exon 1. In the adult human brain, six splicing variants are generated by alternative splicing of exon 2 (light green), exon 3 (blue) and exon 10 (green). All of these variants contain three (3R) or four (4R) microtubule binding repeats (R). Exon 4a (orange) is only transcribed in peripheral nervous system. Exons 6 and 8 (red) are not transcribed in human brain. The microtubule binding repeats constitute the microtubule -binding domain (MTBD). The regions of Tau filaments that have been observed by cryo-EM in Alzheimer’s disease patients (G273/305 to E380) and Pick’s disease patients (K254 to F378 in 3R Tau) are indicated as (AD) and (PD), respectively. (**B**) The 5′ ends of tau mRNA transcripts in Alzheimer’s disease patients determined by 5′ RACE experiments (Huin et al., 2017). The position of the IRES in the 5′ UTR is boxed by a dashed line. The positions of methionine codons 1, 11 and 127 are shown in light green. The positions of the 5′ ends are represented by blue circles. The numbers of nucleotides upstream of the AUG start are indicated by numbers prefixed with the minus sign (−). The numbers of nucleotides downstream of the AUG start are indicated by numbers prefixed with the positive sign (+). The deduced alternative transcription start sites (TSS) found in AD patients are summarized in the bottom of the panel and compared with the TSS found in a normal brain (light green circles).

**Figure 2 biomedicines-10-00241-f002:**
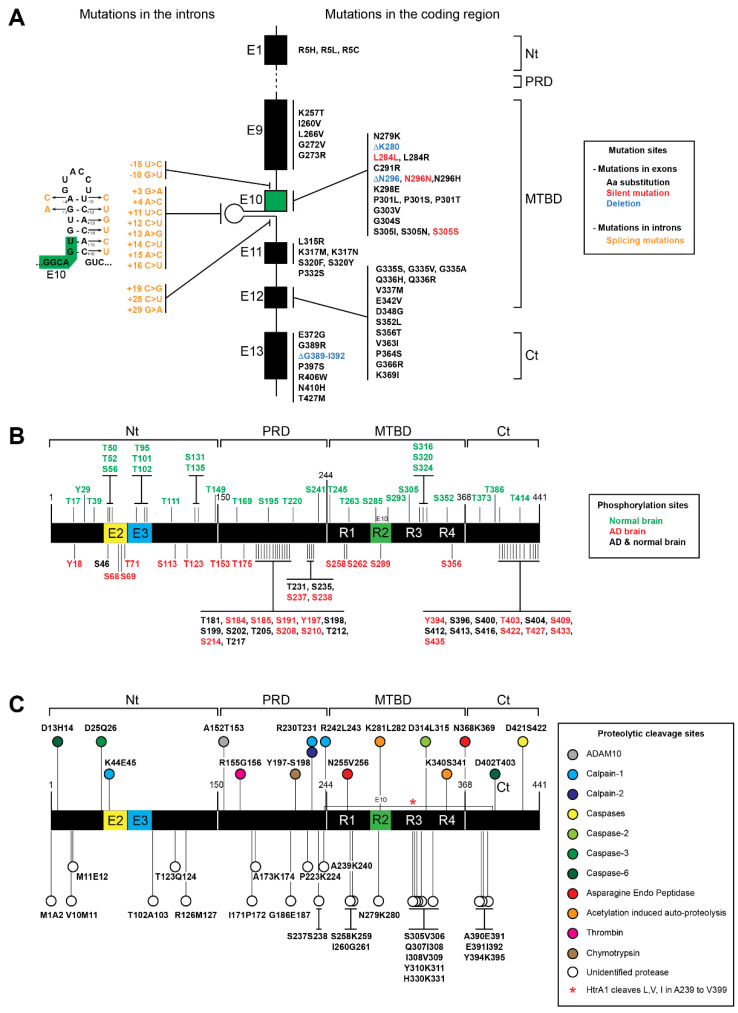
Tau proteins, mutations, phosphorylation and truncation sites. (**A**) Mutations found in tau associated with neurodegenerative diseases. The most frequent mutations are found in the microtubule binding domain (MTBD) and in the C-terminal domain (Ct). Single amino acid substitutions are shown in black, deletions are shown in blue, and silent mutations are shown in red. Several mutations (shown in orange) that affect splicing are also found in the introns flanking exon 10 (shown in green). The mutations are also shown on the intron hairpin that is located close to exon 10 (**B**) The phosphorylation sites that have been found only in normal brain are shown in green above the graphic representing tau protein; phosphorylation sites found only in Alzheimer’s disease (AD) patients are shown in red under the graphic, and phosphorylation sites found in both AD and normal brain are shown in black. (**C**) Cleavage sites by identified proteases are shown above the graphic of tau protein. The color code of the corresponding proteases is indicated in the figure. Other cleavage sites that have been generated by yet unidentified proteases are shown under the graphic and represented by white circles.

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
