# Peer review of "Tau mRNA Metabolism in Neurodegenerative Diseases: A Tangle Journey"

_biomedicines, 2022, doi:10.3390/biomedicines10020241_

Round 1

Reviewer 1 Report

This is a comprehensive assay about the Tau mRNA metabolism.

I do have two minor suggestions:

  • Figure 2A illustrates the MAPT mutations associated with neurodegeneration. Most of these can be divided in two groups missens mutations or splicing mutations. Could this information be added into the illustration with a colour code.
  • Subchapter 2 (Post-translational modification of tau Protein) of chapter tau protein metabolism is opening in the beginning to other post-translational modification like acetylation, methylation, glycation etc. These modifications are later not reviewed at all. Also tau proteolytic truncation is only illustrated in one sentence. This subchapter should be complemented by the authors with further information.

Author Response

Reviewer 1

This is a comprehensive assay about the Tau mRNA metabolism.

I do have two minor suggestions:

  • Figure 2A illustrates the MAPT mutations associated with neurodegeneration. Most of these can be divided in two groups missens mutations or splicing mutations. Could this information be added into the illustration with a colour code.

We have modified Figure 2A accordingly to the reviewer’s comment (the splicing mutations are shown in orange) and updated the corresponding figure legend.

  • Subchapter 2 (Post-translational modification of tau Protein) of chapter tau protein metabolism is opening in the beginning to other post-translational modification like acetylation, methylation, glycation etc. These modifications are later not reviewed at all. Also tau proteolytic truncation is only illustrated in one sentence. This subchapter should be complemented by the authors with further information.

We have added to subchapter 2 brief considerations about acetylation, ubiquitination, methylation and glycosylation of tau. We also added references to reviews (Alquezar et al., 2021; Šimić et al., 2016) that focus on tau PTMs that can be found in the references. We do not want to extend this topic further because we believe that tau protein PTMs are not the main scope of this review.

Concerning Tau proteolytic truncations, we already discussed this topic in details in the subchapter 3 (Tau proteolysis).

Reviewer 2 Report

The review: „Tau mRNA metabolism in neurodegenerative diseases: a tangle journey " by Paulo J da Costa and colleagues elaborately discuss the impact of mRNA metabolism on tauopathies onset and development in light of so far illustrated association of tau on general mRNA metabolism in taupathies. The review is nicely composed with relevant information and data extracted from a large number of published works. The manuscript is quite informative as tau protein are major contributors in several neurodegenerative diseases, including Alzheimer’s disease (AD) and frontotemporal dementia (FTD).

After thoroughly going through the manuscript, I have a couple of comments:

  1. The mechanism by which tau modulates transcription is still enigmatic. However, it is well-established that that tau forms pathological aggregates and fibrils in neurodegenerative diseases. I would suggest the authors to elaborate this point in the manuscript.
  2. Additionally, the nuclear impact of tau at the transcriptional and post-transcriptional level needs to be briefly discussed in the manuscript.

Author Response

Reviewer 2

The review: „Tau mRNA metabolism in neurodegenerative diseases: a tangle journey " by Paulo J da Costa and colleagues elaborately discuss the impact of mRNA metabolism on tauopathies onset and development in light of so far illustrated association of tau on general mRNA metabolism in taupathies. The review is nicely composed with relevant information and data extracted from a large number of published works. The manuscript is quite informative as tau protein are major contributors in several neurodegenerative diseases, including Alzheimer’s disease (AD) and frontotemporal dementia (FTD).

After thoroughly going through the manuscript, I have a couple of comments:

  1. The mechanism by which tau modulates transcription is still enigmatic. However, it is well-established that that tau forms pathological aggregates and fibrils in neurodegenerative diseases. I would suggest the authors to elaborate this point in the manuscript.

We have added few sentences about the hypothetical mechanism by which tau regulates transcription in the chapter “Tau protein and cellular mRNA metabolism”

We added the following sentences:

Although the precise mechanism by which tau modulates transcription is unknown, tau involvement in heterochromatin and DNA methylation are factors to take into consideration. Besides, tau-mediated translocation of known transcription regulators like SFPQ and TDP-43 to the cytoplasm might be of utmost importance. Thus, the tau aggregation into SGs may also be a key factor for tau transcription modulation.

  1. Additionally, the nuclear impact of tau at the transcriptional and post-transcriptional level needs to be briefly discussed in the manuscript.

We have briefly discussed about the nuclear role of tau at the transcriptional and post-transcriptional level in the last paragraph of the chapter “Tau protein and cellular mRNA metabolism”

We added the following paragraph:

Interestingly, it was described that tau also has a nuclear role, being crucial for DNA and RNA integrity in vivo. Furthermore, tau oligomerization triggers the loss of nucleic acid protective effect of monomeric tau [10,23]. Besides, tau has a big impact on the transcription and post-transcription regulation. Indeed, tau nuclear levels and phosphorylation state reduced global synthesis of RNA (including rRNA) and was suggested to be relevant to ribosomal biogenesis and function (Maina et al., 2016; Maina et al., 2018). Moreover, tau was shown to be crucial for ribosomal DNA transcriptional repression by being involved in heterochromatin and DNA methylation (Maina et al., 2018). Also, tau seems to have a role in translocating nuclear RBPs to cytoplasm (Vanderweyde et al., 2012; Vanderweyde et al., 2016). This might be due to tau interaction with nucleoporins impairing the protein transport between the nucleus and the cytoplasm (Eftehkharzadeh et al., 2018). Considering under basal conditions most RBPs are nuclear, and upon stress, they are translocated to the cytoplasm where may trigger the SGs formation (Maziuk et al., 2018; Cruz et al., 2019). The relevance of tau in this process grants tau a pivotal role in global mRNA metabolism. Altogether, the SGs formation and protein aggregation have a huge impact on tauopathies due to their role in transcription regulation, RNA integrity, ribosomal function and subcellular compartmentalization.

Reviewer 3 Report

Tau proteins are known to be mainly involved in regulation of microtubules dynamics. Paulo J da Costa et al. has investigated Tau mRNA metabolism in neurodegenerative diseases: a tangle journey.

In general the manuscript contain relevant paragraphs that have been discussed. The selection of bibliography is appropriate to the content of the manuscript. In the conclusion, the authors have included short thoughts.

The manuscript is very enjoyable to read, but after close evaluation of the paper I suggest revision according to the next point:

1. Indicate the element of novelty.

2. Paragraph 3 (Noncoding RNAs: from microRNAs to long noncoding RNAs) should be corrected: “MicroRNAs (miRNAs) are small RNAs (~22 nts long) that regulate the expression of complementary messenger RNAs and therefore play important posttranscriptional regulatory roles in eukaryotic cells [77]. In humans and other mammals, these small RNAs help modulate the expression of most mRNAs shaping the transcriptome [77]. In brief, miRNAs are transcribed by RNA polymerase II as primary miRNAs (pri-miRNAs). The pri-miRNAs are processed by Drosha/DGRC8 complex in the nucleus before being exported to cytoplasm as a hairpin-shaped precursor miRNA (pre-miRNA). Once in the cytoplasm, Dicer generates the functional ~22nt long double-stranded miRNA [77]. Then, 200 the mature miRNA in association with the RNA-silencing complex (RISC) will encounter the complementary target mRNA leading to its translational repression or degradation. The miRNA-mediated regulation is very versatile since a single miRNA can target several mRNAs and a single mRNA can be under control of different miRNAs [77].”

3. The literature 16, 17, 18, 19, 22 should be corrected.

4. Part "conclusion" should be completed.

5. Please correct some parts of the manuscript. “Giaccone, G.; Rossi, G.; Farina, L.; Marcon, G.; Di Fede, G.; Catania, M.; Morbin, M.; Sacco, L.; Bugiani, O.; Tagliavini, F. Familial frontotemporal dementia associated with the novel MAPT mutation T427M [4]. J. Neurol. 2005, 252, 1543–1545.”

Author Response

Reviewer 3

Tau proteins are known to be mainly involved in regulation of microtubules dynamics. Paulo J da Costa et al. has investigated Tau mRNA metabolism in neurodegenerative diseases: a tangle journey.

In general the manuscript contain relevant paragraphs that have been discussed. The selection of bibliography is appropriate to the content of the manuscript. In the conclusion, the authors have included short thoughts.

The manuscript is very enjoyable to read, but after close evaluation of the paper I suggest revision according to the next point:

  1. Indicate the element of novelty.

The element of novelty in this review is mentioned in the last sentence of the abstract: “Thus, we pretend to portray how mRNA metabolism of, or mediated by tau is associated with neurodegenerative diseases.” It is also indicated in the completed conclusion section.

  1. Paragraph 3 (Noncoding RNAs: from microRNAs to long noncoding RNAs) should be corrected: “MicroRNAs (miRNAs) are small RNAs (~22 nts long) that regulate the expression of complementary messenger RNAs and therefore play important posttranscriptional regulatory roles in eukaryotic cells [77]. In humans and other mammals, these small RNAs help modulate the expression of most mRNAs shaping the transcriptome [77]. In brief, miRNAs are transcribed by RNA polymerase II as primary miRNAs (pri-miRNAs). The pri-miRNAs are processed by Drosha/DGRC8 complex in the nucleus before being exported to cytoplasm as a hairpin-shaped precursor miRNA (pre-miRNA). Once in the cytoplasm, Dicer generates the functional ~22nt long double-stranded miRNA [77]. Then, 200 the mature miRNA in association with the RNA-silencing complex (RISC) will encounter the complementary target mRNA leading to its translational repression or degradation. The miRNA-mediated regulation is very versatile since a single miRNA can target several mRNAs and a single mRNA can be under control of different miRNAs [77].”

We went through this paragraph and found the followings points to correct. We have added the following corrections in this paragraph:

In brief, miRNAs are transcribed by RNA polymerase II as primary miRNAs (pri-miRNAs). The pri-miRNAs are processed by Drosha/DGRC8 complex in the nucleus before being exported to cytoplasm as a hairpin-shaped precursor miRNA (pre-miRNA). Once in the cytoplasm, Dicer generates a ~22nt long miRNA duplex [77]. Then, one strand of miRNA duplex (the mature miRNA) in association with the RNA-silencing complex (RISC) will encounter the complementary target mRNA leading to its translational repression or degradation. ted

  1. The literature 16, 17, 18, 19, 22 should be corrected.

Corrected

  1. Part "conclusion" should be completed.

We have added a whole paragraph in the conclusion part of the manuscript.

  1. Please correct some parts of the manuscript. “Giaccone, G.; Rossi, G.; Farina, L.; Marcon, G.; Di Fede, G.; Catania, M.; Morbin, M.; Sacco, L.; Bugiani, O.; Tagliavini, F. Familial frontotemporal dementia associated with the novel MAPT mutation T427M [4]. J. Neurol. 2005, 252, 1543–1545.”

Corrected
